# Intracellular Detection and Localization of Nanoparticles by Refractive Index Measurement

**DOI:** 10.3390/s21155001

**Published:** 2021-07-23

**Authors:** Alain Géloën, Karyna Isaieva, Mykola Isaiev, Olga Levinson, Emmanuelle Berger, Vladimir Lysenko

**Affiliations:** 1UMR Ecologie Microbienne Lyon (LEM), CNRS 5557, INRAE 1418, Claude Bernard University of Lyon, VetAgro Sup, Research Team “Bacterial Opportunistic Pathogens and Environment” (BPOE), University of Lyon, F-69622 Villeurbanne, France; emmanuelle.danty@univ-lyon1.fr; 2IADI, INSERM, Université de Lorraine, F-54000 Nancy, France; karyna.isaieva@gmail.com; 3LEMTA, CNRS, Université de Lorraine, F-54000 Nancy, France; mykola.isaiev@gmail.com; 4Ray Techniques LTD, P.O. Box 39162, Jerusalem 9139101, Israel; olga.levinson@nanodiamond.co.il; 5Light Matter Institute, UMR-5306, Claude Bernard University of Lyon, 2 rue Victor Grignard, F-69622 Villeurbanne, France; vladimir.lysenko@insa-lyon.fr

**Keywords:** label-free imaging, refractive index, nanoparticle, quantitative image analysis, real-time analysis, single cell analysis, digital holographic microscope

## Abstract

The measuring of nanoparticle toxicity faces an important limitation since it is based on metrics exposure, the concentration at which cells are exposed instead the true concentration inside the cells. In vitro studies of nanomaterials would benefit from the direct measuring of the true intracellular dose of nanoparticles. The objective of the present study was to state whether the intracellular detection of nanodiamonds is possible by measuring the refractive index. Based on optical diffraction tomography of treated live cells, the results show that unlabeled nanoparticles can be detected and localized inside cells. The results were confirmed by fluorescence measurements. Optical diffraction tomography paves the way to measuring the true intracellular concentrations and the localization of nanoparticles which will improve the dose-response paradigm of pharmacology and toxicology in the field of nanomaterials.

## 1. Introduction

Imaging of living cells in their natural environment is essential to study biological processes. To obtain real-time high-resolution images of cells remains a technical challenge because cells are mainly composed of water, giving them physical properties close to the culture medium. Moreover, the three-dimensional spatial arrangement of cell structures is essential for the living processes. Tomographic phase microscopy technics have increasingly been applied to biomedical research [1]. The application of digital holography to microscopy allowed producing a reliable quantitative phase mapping of biological samples [2,3,4]. Recently Cotte et al. [5] developed a three-dimensional computed holographic and tomographic microscope visualizing the 3D morphological structures by contrasting the refractive indices observed via a laser monochromatic wavelength. It is then possible to characterize cells using their refractive indices under label-free conditions. Numerous studies have emphasized the importance of the refractive index of biological materials for fundamental biology and biomedical diagnostics. Indeed, the refractive index is a biophysical parameter correlated with mechanical, electrical, and optical properties of the cells [6,7,8,9,10]. It was suggested that the nuclear refractive index may be a novel physical parameter for diseases, such as early cancer detection [7,11,12]. Nanoparticles have attracted great scientific interest in recent years because of their abundant and versatile applications in medicine including imaging, biosensing, diagnostics, and the therapeutic field. A wide variety of nanoparticles has been produced, although they are finely characterized in size, surface charges and composition, they are immediately bounded by proteins and ions soon after their addition to biological fluids, forming a protein corona preventing the prediction of their biological performance. The protein corona formed is particle-dependent, cell type-dependent, and environment-dependent, resulting in very poor translational predictions from in vitro to in vivo [13]. The toxicity of an NP is measured on proliferating cells in the presence of increasing concentrations of NP. The concentration at which cells stop to proliferate is considered toxic. Suppose an NP that is not at all taken up by cells. It will not be regarded as toxic, it is not wrong, since cells will not be altered by these NPs, except that NPs could have been toxic if they were able to penetrate the cells. There is a confusion between toxicity and uptake. Truly safe NPs are those present inside cells that do not impact cell life. To establish the safety of NPs, their toxicity should be expressed in the function of their intracellular presence and not by the concentration applied on cells. However, it is very difficult to measure the intracellular concentration of nanoparticles. Although computational models of particle sedimentation and diffusion have been developed and provide a good approximation of NPs toxicity in vitro, the gold standard for particle dosimetry for in vitro nanotoxicology studies remains the direct experimental measurement of the cellular content of the particle studied [14]. Considering that nanoparticles may have a refractive index very different from that of cell components, it should, in principle, be possible to localize them at least in the case of accumulation, and/or to measure their impact inside cells. The aim of the present study is to state whether it is possible to detect and localize the presence of unlabeled nanoparticles inside cells, based on measuring their refractive indices.

## 2. Materials and Methods

### 2.1. Measurement of Refractive Indices

The commercial microscope Nanolive (Nanolive SA, Lausanne, Switzerland) was used to measure refractive indices and to reconstruct 3-dimensional refractive index (3-D RI) images of the cells. A green light (520 nm) from a laser diode was split into sample and reference beams. The sample was positioned between a high-numerical-aperture air objective (60× magnification) beneath the sample and a low power laser (0.2 mW/mm^2^) reflected by a rotational illumination arm above. The sample was illuminated with the laser beam inclined at 45° rotating 360° around the sample. A series of holograms was recorded via a digital camera by combining the beam that had passed through the sample with the reference beam. One hundred holograms were captured per rotation. High-resolution images (Δxy = 200 nm; Δz = 400 nm) of each sample layer were created by employing a synthetic aperture and multiple-viewpoint-holographic methods. The software displayed a comprehensible 96 z-stacks cell image in gray scale for a depth of field of 30 µm. A version of the Nanolive microscope was equipped to detect the fluorescence. It was used to detect and localize the presence of fluorescent nanoparticles. 

Raw data were transferred into FIJI, an open-source platform for biological image analysis [15]. Graphs of pixels for each refractive index were constructed to compare experimental versus control conditions. Refractive indices (RI) were measured on one stack. Measurements performed on five stacks showed no significant difference (not shown).

Additionally, it was possible to colorize cells according to their refractive indices. Basic manipulations with image data were implemented using OpenCV library. Pixels ranging from 1.39 to 1.41 have been colorized in red. The number of pixels of each color was measured and used to estimate the uptake of NPs by one cell. 

### 2.2. Cell Culture and Nanoparticle Preparation

Cell line Panc-1 (ATCC CRL1469) was initially grown in flasks containing Dulbecco’s modified Eagle’s medium 4.5 g/L glucose supplemented with 10% newborn calf serum 100 IU penicillin, 100 µg streptomycin, and 0.25 mg/L amphotericin B at 37 °C in a water-saturated atmosphere with 5% CO_2_, in a Heraeus incubator. Cell suspensions were produced using 0.05% trypsin. The cell concentration was measured using a Scepter pipet (Millipore). The cells were seeded at 5000 per well on Ibidi plates (35 mm). After 48 h under control conditions, the cells were either exposed or not exposed to 0.25 or 0.5 mg/mL to nanodiamond nanoparticles for 24 h. These concentrations of nanodiamonds were not toxic to the cells. The cell cultures were rinsed with fresh culture medium before observation.

Nanodiamonds were produced by laser synthesis as previously described [16] by Ray Techniques Ltd. (Givat Ram, Israel) (ref RayND-W-5D), with an average size of 4–5 nanometers possessing a cubic diamond lattice in the core and a shell of graphene-like structure with various functional groups on the surface. 

### 2.3. Cell Toxicity Measurement

Panc-1 cells were exposed for 24 h to ND at 0.25 or 0.5 mg/mL or to the cell death inducer staurosporin at 2 mM (St) as positive control, then for 15 min to propidium iodide (PI) 1 mg/mL at 37 °C, and finally fixed with formalin 3%. Cells were then labeled with Hoechst 33258 in phosphate buffer saline buffer with 0.1% triton. PI is a red-fluorescent cell viability dye which is excluded from live cells with intact membranes, but penetrates into dead or damaged cells. PI intensity was measured at Em/Ex: 535/617 nm on a Cytation 3 plateform (Biotek Instrument, Winooski, VT, USA). Data are presented as mean values of 7 replicates +/−SEM.

### 2.4. Statistical Analysis

The data are presented as mean values ± standard error of the mean (SEM) of at least triplicates. The statistical analysis was performed with StatView 4.5 software for Windows. The data were analyzed using Student’s *t*-test and one way ANOVA followed by Fisher’s protected least significance difference [PLSD], post hoc test.

## 3. Results

### 3.1. Nanodiamond Toxicity

Nuclei counts showed no decrease in response to ND either at 0.25 or 0.5 mg/mL compared to control (Figure 1A). The cell death inducer staurosporine, used as a positive control, resulted in a significant decrease in nuclei number. Dead or damaged cells were identifed using the PI. ND did not increase dead or damaged cells in opposite to staurosporine (Figure 1B). In Figure 1C, left panel shows Hoechst labeling of nuclei, central panel is the phase contrast images and left panel corresponds to merged Hoechst (blue) and PI (red).

### 3.2. Refractive Index Measurement

Nanolive produced 3D images based on the refractive index. For each cell, it was possible to produce a graph showing the number of pixels for each refractive index. To avoid variations of cell sizes, the results have been expressed in % of pixels for each refractive index. Based on these graphs, the cells were characterized by the relative number of pixels corresponding to each refractive index value (Figure 2). The RI of control Panc-1 cells varied from 1.365 to 1.395. The maximum values of the refractive index were centered around 1.3725. In controls, 99.5% of the refractive indices were between 1.365 and 1.39. The shape of the curve is slightly asymmetric since values lower than the maximum RI (=1.3725) represented 34% of the pixels. Incubation of Panc-1 cells with 0.25 mg/mL nanodiamonds slightly decreased the maximal value and shifted the curve number of pixels per RI to the right (Figure 2). A higher concentration of nanodiamonds (0.5 mg/mL) amplified both the decrease of the maximal value and the shift to the right. The RI shifts occurring from 1.38 to 1.39 attest to the presence of NDs inside the cells. (Figure 2)

The fluorescence of cells exposed to nanodiamonds was significantly increased compared to those of the control cells. For the control cells the values were 5.1 ± 0.2 arbitrary units (UA), 10.1 ± 0.4 (*p* < 0.05) for cells exposed to 0.25 mg/mL, and 15.1 ± 0.6 (*p* < 0.05) for cells exposed to 0.5 mg/mL (*n* = from 67 to 126).

Knowing the effect of NDs on RI values (Figure 2), the location of nanodiamonds corresponds to the pixels with the highest values. It was then possible to localize them. Since NDs are fluorescent, it was then also possible to localize them directly inside the cells. The comparison of the RI values shifted, and the fluorescence shows a perfect match, demonstrating that the increased RI values correspond to the place where the fluorescence is located, proving that NDs are present inside the cells. Nanodiamonds are localized inside the cytosol around the Golgi apparatus. (Figure 3C). It cannot be excluded that the RI values of some intracellular components may be the same RI values as those of the nanodiamonds. This is the case of chromatin which may reach RI values higher than 1.39. In the present case, NDs did not penetrate the nucleus, since they are not fluorescent (Figure 3C).

## 4. Discussion

ND exerted no toxicity at the concentrations tested (Figure 1). An imprint of cells can be produced using a tomographic holographic microscope by reporting the number of pixels for each refractive index value (Figure 2). Although the cell lines show some variations in the shape of the curve, the refractive index of Panc-1 cells remains in a well-defined range between 1.365 and 1.395. These values are in accordance with data from the literature [9]. 

When incubated in the presence of nanoparticles, the frequency of some refractive indices increased in comparison to those observed in the control cells. That is true for the highest values of RI, (RI > 1.39) (Figure 2). It allowed the identification of cells exposed to nanoparticles. One of the most interesting characterizations of the refractive index is that it does not require labeled NPs. With nanoparticles being too small to be directly observed, it is nevertheless possible to observe their intracellular accumulation. Based on the number of pixels for each refractive index per cell, it is possible to localize NPs inside cells. Moreover, the spreading of NPs inside cells without any accumulation sites cannot be excluded. Although accumulation most likely represents the majority of NPs taken up by cells, it cannot be excluded that NPs might also induce structural changes that may modify the refractive indices within cells. Nevertheless, one may expect that such changes of the RI would remain within the range of the RI of biological values.

Although the shift of the RI is not an absolute concentration of NPs, it is possible to compare NP uptake by cells in the presence of increasing concentrations (Figure 2). The shifted values of refractive indices can be localized inside cells, showing that nanodiamonds are located in the cytoplasm, mostly around the endoplasmic reticulum and do not enter the nucleus. In the present case, the use of fluorescent nanodiamonds also allowed their localization within cells. It has been confirmed that the highest refractive index values correspond to the fluorescence that validates the measurement of refractive indices for the detection of nanoparticles (Figure 3C). 

One limitation of the present method is the refractive index of the nanoparticles studied. NP detection will be improved for a refractive index much higher than those observed inside the cells (1.365–1.395). Nanoparticles with an RI in the range of those of the cell components would most likely remain invisible, although some changes in the shape of the repartition curve of the refractive index could eventually be observed. Furthermore, the quantification of NPs also depends on their intracellular behavior. Indeed, a strong concentration inside vesicles would be more easily detectable than NPs homogeneously dispersed within the cells. An important advantage of the present method is that cells can be observed live without any preparation. Since the measurement is performed in 3D, it is theoretically not necessary to wash out the nanoparticles from the culture medium.

The transfer of the raw data to the FIJI software is not automatic, it is time consuming as the delimitation of cells is required for the integration of the refractive indices. Nevertheless, it provides the opportunity to dissect the cell into its different components. It is indeed possible to select the nucleus or only the cytoplasm to measure the evolution of cell indices and to study a specific localization of nanoparticles.

The possibility of measuring nanoparticle accumulations within the cells allowed studying the kinetics as well as the uptake mechanisms and the possible release of these NPs over time. A previous study [17] also pointed out the feasibility to measure the intracellular localization of NPs based on their refractive index. More recently [18] used a Nanolive microscope to analyze the uptake of nanodiamonds. The cells were exposed to NDs for a long time, 2 or 7 days, showing a decrease in intracellular content over time, nevertheless the authors did not consider the possibility to quantify nanoparticles using the refractive index. The main advantage of measuring the RI is that there is no sample preparation. Cells are observed under living conditions without any treatment. The application fields of refractive index measurements are wide, starting from measuring the true relationship between cell toxicity and the intracellular concentration of nanoparticles to the kinetics of NP uptake and their intracellular localization. Using the RI renders the labeling of NPs unnecessary, any type of nanoparticles can be studied as long as its refractive index is significantly different from that of the cell components.

## 5. Conclusions

The present study demonstrates that it is possible to localize nanoparticles inside cells based on the changes of refractive index. The efficiency of the measure depends on the size of nanoparticles, their refractive index, their behavior inside cells. The generalization of such measure should be done with care, compared with the appropriate control. It will help to identify true non-toxic nanoparticles, which are nanoparticles present inside cells which produce no deleterious effect on these cells.

## Figures and Tables

**Figure 1 sensors-21-05001-f001:**
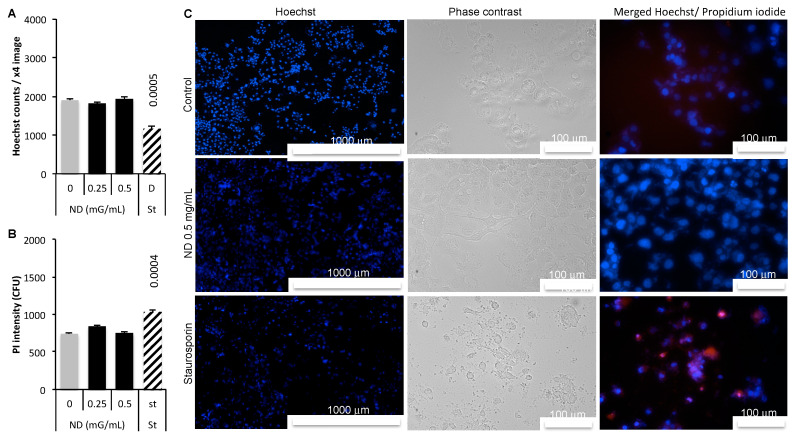
Panc-1 cells treated during 24 h with 0.25 or 0.5 mg/mL ND (0 mg/mL = control) or with cell death inducer Staurosporin 2 mM (St) as positive control, then 15 min with propidium iodide (PI) 1 mg/mL at 37 °C, and finally fixed with formalin 3% then labeled with Hoechst 33258 in phosphate buffer saline buffer with 0.1% triton. (**A**) Nuclei counts were obtained from Hoechst counts (**B**) PI intensity was measured at Em/Ex 535/617 nm on a Cytation 3 plateform (Biotek Instrument, Winooski, VT, USA). Data are presented as mean values of 7 replicates +/−SEM with significant difference at *p* < 0.05. (**C**) left panel corresponds to nuclei coloration with Hoechst (blue), ×4, central panel, representative ×20 phase contrast and right panel shows the merged Hoechst (blue) and PI (red) images.

**Figure 2 sensors-21-05001-f002:**
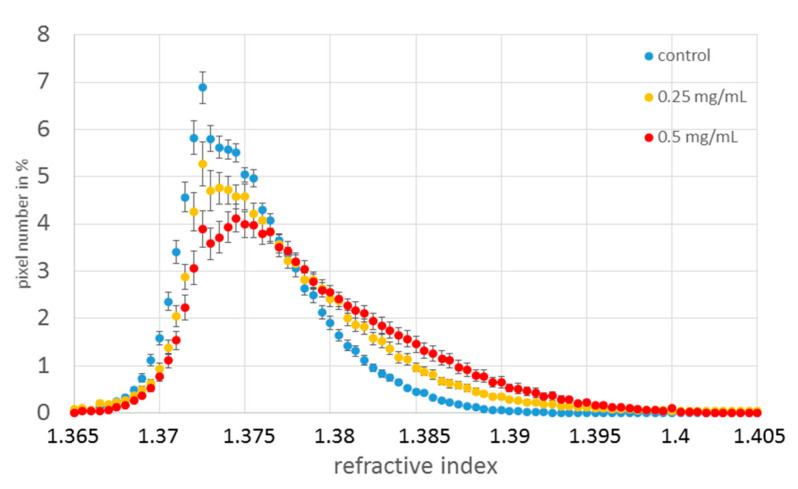
Repartition of refractive indices of human pancreatic cancer cells (Panc-1, *n* = 60), an effect of a 24-h incubation in the presence of nanodiamonds (0.25 or 0.5 mg/mL) on Panc-1 cells (*n* = 60). The controls have not been exposed to NPs. The results are expressed in % of pixel numbers to avoid the effects of cell size.

**Figure 3 sensors-21-05001-f003:**
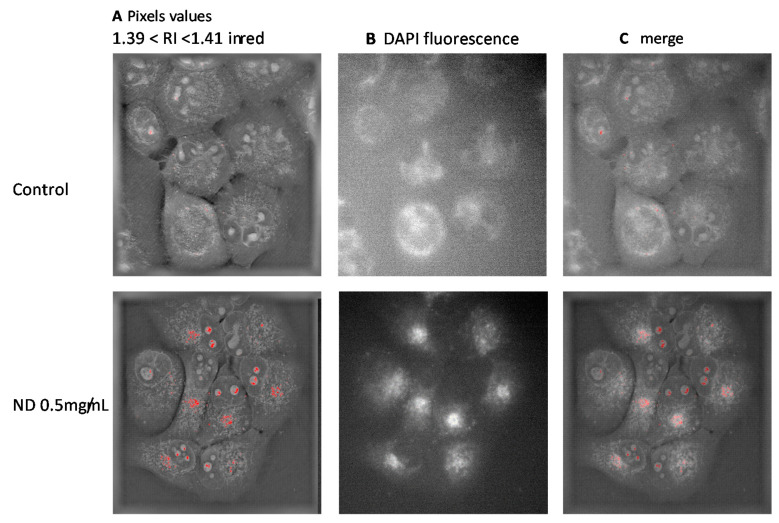
(**A**) Images based on the refractive indices of cells. Refractive indices between 1.39 and 1.41 are colorized in red. (**B**) Fluorescence images of cells. (**C**) Merged image of RI and DAPI fluorescence (0.5 mg/mL). The absence of fluorescence means that there is no nanoparticle inside the nuclei. There is a perfect match between the shifted RI and the fluorescence.

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
