# Peer review of "Intracellular Detection and Localization of Nanoparticles by Refractive Index Measurement"

_sensors, 2021, doi:10.3390/s21155001_

Round 1
Reviewer 1 Report
This study presents whether the intracellular detection of nanodiamonds is possible from the measure of refractive index. Based on optical diffraction tomography of treated live cells, results show that unlabeled nanoparticles can be detected and localized inside cells.
Thank you very much for your good work. My review points the followings,
You need to double-check the scientific writing of your manuscript, take care of punctuations and make shorther and clearer sentences. The English needs to be imporved.
Line 23: Instead of [reviewed in 1], just [1] is enough.
Line 39: “Numerous studies have emphasize the importance of refractive index of biological materials for fundamental biology and biomedical diagnostics”: Explain why?
Line 40: There should be a space: diagnostics [6-10].
Line 47: their addition in biological fluids could be their injection into biological fluids...
Line 48: Explain first what is protein corona
Line 50-51: Rewrite the sentence to make it more clear. You can make it two sentences.
Line 52: You need a comma: On the opposite, or a better suggestion can be: Contrary,..
Line 52-53: Rewrite the sentence, again the meaning is not clear and it is difficult to follow the sentence, make it two sentences. Please use shorter sentences.
Line 57: The difficulty lies in the fact that it is extremely difficult: Make it a better sentence.
Line 63: It should be possible to localize …
Line 64-65: Rewrite the sentence.
Line 69: Measureïƒ Measurement …
The whole text should be correct for its English.
Line 71: 3D-RI: First write the full name and then use the abbreviations, RI? Explain it here instead of the Line 86.
Materials and methods: Provide the schematic for your experimental setup.
Line 78: There should be space between the numbers and the units: Δxy = 200nmïƒ 200 nm.
Line 84: Raw data was (instead of were), what is FIJI, a software? If yes: give the details about the software, use the reference [15] at the end of the sentence.
Line 90: in case their value (i.e. RI) lies in user-defined range: Which value?
Line 92: According to your text, the title should be: Cell culture and nanoparticle preparation (not treatment)
Line 93: Is Cell line Panc01 obtained from ATCC, explain from where you get it.
Line 104-105: which functional groups should be detailly written here.
Figure 1 and especially Figure 2 should be professionally prepared.
Is it Panc 01 or Panc01?
In Figure 1, is control 0 mg/ml? it should be written in the figure legend. Y-axis’s caption is not readable.
Why did you choose the concentration values of the ND’s 0.25 mg/ml and 0.5 mg/ml? It should be explained.
For figure 2, could you perform these experiments with a range of ND concentration and obtain the correlation curve between the dose and fluorescence intensity?
Line 142-144: Knowing the effect of NDs on RI values, it was then possible to localize the nanodiamonds inside cells. Since ND are fluorescent, it was then also possible to localize them directly inside cells.
When there are aggregates of NDs, how do you find the precise location within the cell
Is it possible to have a merged image of NDs and DAPI to identify the localization of the NDs?
Figure 3 legend needs to be corrected: ib=ndexes between 1.39 and 1.41 are colored in
As stated in the discussion, the control experiments should include the distribution of NDs in the other cell lines or another pancreatic cell line and compare the resolution of RIs.
Another control experiment should present the viability of the cells using the 0.25 mg/ml and 0.5 mg/ml concentration of NDs, is it stable for 24 h? When the cells dye, how the RI changes?
Reviewer 2 Report
Review: Major Revision
Summary:
This manuscript described a refractive index measurement for detection and localization of nanodiamonds (NDs) in live cells. The idea of this work is in some extent of novel. However, the experiment are not enough to support the conclusion and the results are badly described. And the manuscript is poorly written and need a thorough revision.
Major Review:
1. In figure 1, can the authors compare the measured curve of pixel number of NDs-only solution (in a similar volume with a cell, but not inside the cell)? For example, the curve can be measured under 3 different concentrations: 0 mg/ml, 0.25 mg/ml, 0.5 mg/ml.
2. Similarly, can the author provide the fluorescence intensity measured for the NDs-only solution (outside of the cell) as comparison?
3. The authors mentioned that the nanoparticles toxicity inside a cell is difficult to measure. However, the experimental results and description have no relationship with cell toxicity at all. Maybe the authors can adjust the contents of the manuscript either providing some experiments for cell toxicity or revising the narrative in the introduction to be more suitable to the experiments results.
4. Can the authors provide a much more thorough description in the results section?
5. Can the authors provide a 3D image of the cell to demonstrate that regions that could be localization of nanodiamonds?
Minor Review:
1. Page 3 (Section 2.3):
Data are presented as mean values ± SEM of at least triplicates.
=> Please provide the full name for any abbreviation (e.g., SEM) in the manuscript.
2. Page 3 (Section 3.1):
Although the all system produces high quality images with a high resolution (200 nm in x,y), it is not possible to collect all the RI values for only a single cell.
=> Although all the system ...
3. Page 5:
Figure 3
=> Can the author add a title for the figure caption?
4. Page 4 and 5:
Both titles of 3.2 and 3.3 are "Fluorescence measurement."
Reviewer 3 Report
This work presents one approach to estimate the accumulation of nanoparticles inside the cells. This makes sense for designing nanoparticle labelling based study, which has been widely used in cell imaging (Nat. Photonics 13, 480–487 (2019)). Although it is interesting, it still needs more details for easier reading. I will list my concerns point by point.
1, The authors make some adjustment on the commercial microscopy to achieve holographs. These adjustments should be presented. A schematic of system should be added in the article. In addition, the numerical aperture of objective and camera frame rate should be presented.
2, ND shows in the results section. I guess it represents the NP (nanoparticles). Please confirm it.
3, For refractive index based detection, the noise sources should be discussed. Plot the noise against average period will help you find that the noise mainly comes from shot noise or mechanical noise (Science 2018: Vol. 360, Issue 6387, pp. 423-427; Nat Methods 17, 1010–1017 (2020)). This will help you figure out the next improvement (ACS Photonics 2017, 4, 2, 211–216). I advise the authors to analyze the noise source in their system and add it into discussion about next work.
Round 2
Reviewer 1 Report
Although the time is limited for the revision, since it is a communication paper, it should be prompt and precise. Toxicity of nanoparticles should be shown by live-dead staining and in addition to microplate reading CFU assay also should be provided.
Reviewer 2 Report
I appreciate that the authors take time and effort to consider my suggestions in the previous round of review. I only have a few questions for the updated manuscript.
Minor Review:
1. Page 3:
The authors removed the original titles for section 3.2 and 3.3. However, it is a little weird that the authors kept the title of section 3.1 while the manuscript only have one section in the results section.
2. Page 5:
Since the authors removed the original figure 2, maybe the number of figure 3 should be updated as figure 2 (both in the figure caption and in the text).
Author Response
Answers to referee 2:
The referee is right.
Page 3:
The authors removed the original titles for section 3.2 and 3.3. However, it is a little weird that the authors kept the title of section 3.1 while the manuscript only have one section in the results section.
A new section on ND toxicity has been added in the revised version. The section “Results” has now two subtitles:
3.1 Nanodiamond toxicity
3.2 Refractive index measurement.
- Page 5:
Since the authors removed the original figure 2, maybe the number of figure 3 should be updated as figure 2 (both in the figure caption and in the text).
A new figure (Figure 1) has been added. Figure 3 remains Figure 3 in the revised version.
We acknowledge the referee for his careful reading and his valuable comments.
